# The Angiopoietin Signaling Pathway Is Involved in Inflammatory Processes in Hospitalized COVID-19 Patients

**DOI:** 10.3390/microorganisms11122940

**Published:** 2023-12-08

**Authors:** Rachelle P. Mendoza, Amir Momeni, Nayanendu Saha, Juwairiya Arshi, Elmer C. Gabutan, Nichole Alejandro, Alejandro Zuretti, Prem K. Premsrirut, Dimitar B. Nikolov

**Affiliations:** 1Department of Pathology, University of Rochester Medical Center, 601 Elmwood Avenue, Rochester, NY 14642, USA; juwairiya_arshi@urmc.rochester.edu; 2Department of Pathology, Memorial Sloan Kettering Cancer Center, 1275 York Avenue, New York, NY 10065, USA; momeniba@mskcc.org; 3Structural Biology Program, Memorial Sloan Kettering Cancer Center, 1275 York Avenue, New York, NY 10065, USA; sahan@mskcc.org; 4Department of Pathology, SUNY Downstate Health Sciences University, 450 Clarkson Avenue, Brooklyn, NY 11203, USA; elmer.gabutan@downstate.edu (E.C.G.); alejandro.zuretti@downstate.edu (A.Z.); 5Bouvé College of Health Sciences, Northeastern University, 360 Huntington Avenue, Boston, MA 02115, USA; alejandro.nichole@gmail.com; 6Department of Cell Biology, SUNY Downstate Health Sciences University, 450 Clarkson Avenue, Brooklyn, NY 11203, USA; prem@mirimus.com; 7Mirimus Inc., 760 Parkside Ave, Brooklyn, NY 11226, USA

**Keywords:** COVID-19, angiopoietin, COVID-19-related vasculitis

## Abstract

The viral agent SARS-CoV-2 clearly affects several organ systems, including the cardiovascular system. Angiopoietins are involved in vascular integrity and angiogenesis. Angiopoietin-1 (Ang1) promotes vessel stabilization, while angiopoietin-2 (Ang2), which is usually expressed at low levels, is significantly elevated in inflammatory and angiogenic conditions. Interleukin-6 (IL-6) is known to induce defective angiogenesis via the activation of the Ang2 pathway. Vasculitis and vasculopathy are some of the defining features of moderate to severe COVID-19-associated systemic disease. We investigated the serum levels of angiopoietins, as well as interleukin-6 levels and anti-SARS-CoV2 IgG titers, in hospitalized COVID-19 patients across disease severity and healthy controls. Ang2 levels were elevated in COVID-19 patients across all severity compared to healthy controls, while Ang1 levels were decreased. The patients with adverse outcomes (death and/or prolonged hospitalization) had relatively lower and stable Ang1 levels but continuously elevated Ang2 levels, while those who had no adverse outcomes had increasing levels of both Ang1 and Ang2, followed by a decrease in both. These results suggest that the dynamic levels of Ang1 and Ang2 during the clinical course may predict adverse outcomes in COVID-19 patients. Ang1 seems to play an important role in controlling Ang2-related inflammatory mechanisms in COVID-19 patients. IL-6 and anti-SARS-CoV2 spike protein IgG levels were significantly elevated in patients with severe disease. Our findings represent an informative pilot assessment into the role of the angiopoietin signaling pathway in the inflammatory response in COVID-19.

## 1. Introduction

As COVID-19 infections continue, a better understanding of the pathophysiology of the disease has come to light. The viral agent SARS-CoV-2 clearly affects several organ systems, including the cardiovascular system. The involvement of the cardiovascular system is either due to direct viral tropism or indirectly through systemic-wide inflammation. The inflammation of the blood vessel wall leading to rupture commonly known as vasculitis and the blockage of arterial blood flow or vasculopathy are the outcomes of moderate to severe COVID-19 infections. A comprehensive understanding of the molecular mechanism underlying vasculitis should spur better treatment strategies for COVID-19 patients.

In the light of the observation that vasculitis and vasculopathy pose a major threat during COVID-19 infections, we set out to evaluate the patients’ plasma levels of some of the key secreted signaling molecules involved in angiogenesis, including angiopoietin-1 (Ang1), angiopoietin-2 (Ang2) [1], as well as the cytokine interleukin-6 (IL-6) [2] that causes vascular repercussions. The general aim of this study was to identify potential molecular targets for therapeutic intervention during COVID-19 disease progression. To that end, we conducted ELISA-based measurements of the levels of Ang1, Ang2, IL-6, as well as titers of IgG against the spike protein RBD, in the serum samples of hospitalized COVID-19 patients with various degrees of disease severity.

Tie2 tyrosine kinase receptor signaling in endothelial cells plays a central role in wound healing, vascular integrity, and angiogenesis [3,4]. Ang1, the activating ligand for Tie2, is essential for blood vessel growth and maturation [5]. Ang1 promotes vessel stabilization and enlargement without inducing vascular sprouting [1,6,7]. Vessels exposed to Ang1 are resistant to leak induced by the vascular endothelial growth factor (VEGF) or inflammatory agents [6] suggesting the importance of Ang1-Tie2 signaling in maintaining a quiescent, well-functioning vasculature. Ang2, initially identified as an Ang1 sequence homologue, is expressed predominantly in endothelial cells and smooth muscle cells [8]. Unlike Ang1, the expression of Ang2 is strictly regulated [9]. Its expression is usually low in quiescent mature vessels, but it is significantly elevated in many inflammatory and angiogenic conditions [9]. Ang2 acts in combination with VEGF to initiate angiogenesis. Ang2 is found to be a natural inhibitor of Ang1 [10,11]. A high level of Ang2 was seen in mice models of inflamed pulmonary endothelial cells, and increased Ang2 was associated with a progressive loss of endothelial cell quiescent and junction proteins [12].

SARS-CoV-2 has an increased tendency to bind to endothelial cells with resulting inflammation [13]. An accumulation of inflammatory cells and viral inclusions occur within the vascular endothelium of the heart, small bowel, kidneys, and lungs [14]. Increased cytokines and other pro-inflammatory factors, such as TNF, IL-1, IL-6, and IL-8, are believed to contribute to endothelial injury and subsequent pro-thrombotic events in COVID-19 [15]. A number of co-morbidities including advanced age, diabetes mellitus, hypertension, and obesity, have been linked to vascular dysfunction and altered endothelial cell metabolism [16]. In a study carried out to explore biomarkers in hospitalized COVID-19 patients, Ang2 was found to be a relevant predictive factor for intensive care unit (ICU) admission [17]. On the basis of the increased levels of Ang2 in respiratory distress syndrome, an investigational anti-Ang2 monoclonal antibody (LY3127804) for COVID-19 is now in a Phase 2 clinical trial. However, further investigation of the Ang1 and Ang2 signaling pathway has not been reported in COVID-19 patients. This study aims to investigate the levels of Ang1 and Ang2 in COVID-19 patients during the course of hospitalization, and correlate the levels according to the severity of illness, anti-SARS-CoV-2 IgG titers, interleukin IL-6 level, and clinical outcome.

## 2. Materials and Methods

### 2.1. Patient Samples

De-identified samples from forty-nine (49) randomly selected hospitalized COVID-19 patients in SUNY Downstate University Hospital of Brooklyn, New York, were included in the study (IRB protocol number: 1587443, date of approval: 10 April 2020). All patients were diagnosed and admitted within the first few weeks of the pandemic (March to May 2020). All were presumed to have been infected by the original Wuhan strain (WA1/2020). Multiple serum samples were collected on days 1, 3, 7, 14, 21, and 28, with day 1 being the day of admission. Limited demographics and clinical information were also collected. The clinical status is categorized into two levels of severity: non-severe—patients who were hospitalized but did not require any invasive oxygen support (i.e., intubation) until discharge; severe—patients who were hospitalized and required any invasive oxygen support until discharge or death. Serum samples from three COVID-19-negative, healthy donors were utilized for baseline comparison. A total of 10 to 15 mL of whole blood samples was collected from the antecubital veins of the patients directly into serum separator vacutainer tubes. The blood was allowed to clot at room temperature for 15–30 min before centrifuging at 10,000 rpm for 10 min. Serum was aspirated into 1.5-mL cryotubes and frozen at −80 °C until further analysis.

### 2.2. Characteristics of Patients

Table 1 shows the baseline characteristics of the 49 patients included in this study. Thirty-six patients were African American (73.5%), eight were white (16.3%), one was Asian (2.0%), and four (8.2%) did not reveal this information. The median age was 63.0 years (interquartile range [IQR], 57–70 years). Almost all patients had at least one co-morbidity (48/49, 98.0%). The most common co-existing illness was hypertension (38/49, 77.6%), followed by diabetes mellitus (32/49, 65.3%). The median length of hospitalization was 20 days (IQR 9–28 days), and 12 of the patients died (mortality 24.5%).

### 2.3. Assays: Ang1, Ang2, and IL-6

The levels of soluble Ang1, Ang2, and IL-6 were determined using a commercially available sandwich ELISA kit (R & D Systems, Minneapolis, MN, USA), following the manufacturer’s protocol and as previously described [18]. Fifty microliters of serum sample were diluted with 50 μL of the kit’s sample diluent and added to pre-coated 96-well plate wells. One hundred microliters of serially diluted standards were also added to designated wells. Samples and standards were incubated at room temperature for 2 h. After washing three times with phosphate-buffered saline (PBS) with 0.05% tween solution, 100 μL of the kit’s Detection Antibody was added to each well, followed by incubation for 2 h at room temperature. After washing again with the same wash solution three times, 100 μL of streptavidin-horse radish peroxidase (HRP) was added to each well, followed by incubation for 20 min at room temperature. The wells were again washed three times. One hundred microliters of a color substrate (TMB) were added to each well and incubated for another 20 min at room temperature until the development of a yellow color. Fifty microliters of 2N H_2_SO4 was applied to each well as a stop solution, and absorbance was recorded at 450 nm. The optical density was proportional to the amount of Ang1, Ang2, and IL-6 in the sample, and the concentration was determined by interpolating the optical density in the standard curve using CurveExpert version 2.5.6. Using this ELISA kit, the levels of Ang2 in human plasma measured and reported across a spectrum of studies, are reassuringly consistent. Plasma from healthy controls contains levels of Ang2 of approximately 1–3 ng/mL, and rarely more than 4 ng/mL, whereas plasma from disease shows a range of Ang2 levels from 5 to 10 ng/mL and as high as 20 ng/mL or more.

### 2.4. ELISA IgG Titer

All plasma/serum samples were tested for levels of IgG antibody against the SARS-CoV-2 S1 spike protein receptor RBD (Wuhan strain, WA1/2020) as previously described [18]. Assay controls and serum samples were diluted to 1:100, followed by 2-fold dilution out to 1:102,400, and added to a 96-wells microtiter plate (Thermo Scientific Immulon, Waltham, MA, USA) that was coated with SARS-CoV-2 recombinant RBD (GenScript Biotech, Piscataway, NJ, USA). A horseradish peroxidase-conjugated secondary anti-human IgG (Fab specific) antibody was added to each well to form a specific complex of antigen–antibody bound to the plate surface. The binding reaction was then enhanced visually with SIGMAFASTTM OPD (Sigma-Aldrich, St. Louis, MO, USA) substrate generating a yellow color for positive specimens. After the application of the stop solution (3 M Hydrochloric acid), the color changed from yellow to orange and optical density was measured at 490 nm. When the absorbance value was greater than the cut-off value (OD490 = 0.15), the specimens were reported as a positive result and the resulting titer was reported.

### 2.5. Statistical Analysis

Statistical analyses were carried out with the NCSS 2020 Statistical Software (NCSS, LLC. Kaysville, UT, USA) and R. The outcome variables included disease severity (as determined by a need for ICU admission, a need for intubation, and/or a need for supplemental oxygen), and outcome (early discharge, late discharge (>30 days of hospitalization) and death). Averages were determined for patients across multiple measurements performed at predetermined intervals with overall average, and weekly averages used in comparisons in addition to moving averages to determine relevant time trends. The Mann–Whitney U test was used for two group comparisons of quantitative measures. Comparisons of more than two groups were performed using ANOVA with post hoc analysis using the Tukey test. Associations of categorical variables were evaluated using the chi-square test, Fisher’s exact test, or Spearman’s r. Linear regression analysis was used to investigate correlations between quantitative variables. Alpha was set at 0.05 with two-sided *p*-values less than the alpha considered as significant for single comparisons, with Bonferroni adjustments made when multiple comparisons were made.

### 2.6. Study Approval

The study protocol and other related documents were submitted for review to the SUNY Downstate Medical Center Institutional Review Board prior to implementation. Approval was secured on 10 April 2020 (protocol number: 1587443).

## 3. Results

### 3.1. Ang2 Serum Levels Are Significantly Elevated in COVID-19 Patients

When compared to healthy controls (549.9 pg/mL), Ang2 serum levels were elevated in COVID-19 patients (severe and non-severe). The median Ang2 levels in the three groups were as follows: (i) the non-severe group, 3694.56 pg/mL, IQR: 1669.5–4955.9 pg/mL; (ii) the severe group, 2983.6 pg/mL, IQR: 1294.4–4676.5 pg/mL; and (iii) the healthy group, 549.9 pg/mL, IQR: 454.6–875.2 pg/mL (Figure 1A). The difference between COVID-19 patients (severe and non-severe) and the healthy control was statistically significant (Mann–Whitney U *P* = 0.0001). The Ang2 levels were also elevated in COVID-19 patients with heart disease, who either died from COVID-19-related complications (ANOVA *P* = 0.0001) or had adverse clinical outcomes (either death or prolonged hospitalization (>30 days); ANOVA *P* = 0.001). This finding is in agreement with a previous study with COVID-19 patients (Smadja et al., 2020 [17]), where elevated levels of Ang2 were associated with more severe disease requiring intensive care. Furthermore, a visual inspection of the time trend changes in Ang2 levels in patients revealed the following: (i) for patients who survived, the levels of Ang2 remained relatively stable throughout hospitalization, while (ii) for patients who eventually died, they continued to have an increasing trend of Ang2 even beyond the second week of illness (Figure 1B).

### 3.2. Ang1 Serum Levels Are Decreased in COVID-19 Patients

The median Ang1 levels in COVID-19 patients (severe and non-severe) was mildly decreased with a median of 18,265.3 pg/mL (IQR: 12,533–25,660), while in healthy controls, the median Ang1 levels was 29,441.2 pg/mL (IQR: 25,953.5–32,703.7) (Mann–Whitney U *P* = 0.075). Ang1 levels did not differ between severe and non-severe patient groups (Mann–Whitney U *P* = 0.114; Figure 2A). A visual inspection of the time-course progression of Ang1 levels showed an increasing trend on day 7 of illness for patients who survived and were stable compared to patients who eventually died (Figure 2B).

### 3.3. The Dynamic Levels of Ang1 and Ang2 during the Clinical Course May Predict Adverse Outcomes in COVID-19 Patients

The median Ang2-to-Ang1 ratio was also elevated in COVID-19 patients (severe and non-severe) compared to healthy controls (median: 0.18 in non-severe versus 0.12 in severe, ANOVA *P* = 0.042; Figure 3A). We tracked the moving patient averages of Ang2 and Ang1 levels between patients with adverse outcomes and those without. Interestingly, patients who had no adverse outcomes had increasing levels of both Ang1 and Ang2 followed by a decrease in both, while those patients with adverse outcomes had relatively lower and stable Ang1 levels but elevated Ang2 levels (Figure 3B). This may suggest an important role of Ang1 in modulating the Ang2-related inflammatory mechanisms in COVID-19 patients.

### 3.4. IL-6 Serum Levels and Anti-SARS-CoV2 IgG Titers Are Significantly Elevated in COVID-19 Patients

The serum levels of IL-6 were significantly elevated in COVID-19 patients across disease severity (median: 55.2 pg/mL in COVID-19 versus 0.0 pg/mL in healthy patients, Mann–Whitney U *P* = 0.001; Figure 4A). A visual inspection of the IL-6 trends shows that in patients who died from COVID-19 complications, IL-6 levels continued to rise during the first two week of illness and remained elevated until the third week (Figure 4B). In patients who survived, IL-6 levels were observed to decrease after the first week of illness.

The amount of IgG antibodies against the SARS-CoV-2 spike protein was significantly elevated in COVID-19 patients with severe disease (median: 27,200 antibody titer in severe versus 6525 in non-severe, Mann–Whitney U *P* = 0.0001; Figure 5). However, IgG antibodies levels did not seem to correlate with clinical outcomes.

## 4. Discussion

In this study, the dynamic levels of angiopoietins were investigated in hospitalized COVID-19 patients during the first few weeks of the pandemic. The angiopoietin levels were correlated with disease severity, disease outcomes, and the levels of serum IL-6 and IgG.

We observed that all patients, regardless of disease severity, had significantly elevated serum levels of Ang2. This indicates the involvement of the Ang2 pathway in the systemic inflammatory pathogenesis of COVID-19, where endotheliopathy has been identified as a definite mechanism. Ang2 may be involved in the destabilization of blood vessels following SARS-CoV-2 infection or as a response to elevated levels of cytokines and other pro-inflammatory factors post infection [17,19].

Our studies showed that an elevated Ang1 level was associated with favorable outcomes in COVID-19 patients. This may be attributed to the anti-inflammatory and vascular stabilizing activities of Ang1. Ang1 binds to Tie2 and suppresses the activation of endothelial cells by inhibiting vascular cell adhesion molecules and procoagulant tissue factor expression [20,21]. Uncontrolled inflammation has been linked to poor prognosis in critically ill COVID-19 patients. SARS-CoV-2 is known to directly infect vascular endothelial cells and induce immune and thromboinflammatory pathways that lead to coagulopathy, endotheliopathy, and vasculitis [22,23]. The mechanism of Ang1 in neutralizing these fatal inflammatory effects in COVID-19 must be further investigated.

The elevated concentration of Ang2 relative to Ang1 may be utilized as a biologic marker of prognosis among COVID-19 patients. Our results are consistent with a study on the role of Ang2 and Ang1 levels as predictive biomarkers of mortality in patients with acute lung injury [24]. The researchers found that an elevated Ang2 concentration relative to Ang1 (Ang2-to-Ang1 ratio) was associated with an increased risk of death, especially among those with increased endothelial activation as measured by pulmonary dead space fraction. Acute lung injury is a known complication of severe COVID-19; therefore, an elevated Ang2-to-Ang1 ratio can be expected to be associated with poor clinical outcomes. Further investigation are warranted to confirm the utility of this ratio to predict the severity of lung injury among COVID-19 patients.

In accordance with published findings, in our study, the IL-6 and IgG antibody levels were higher in patients with severe clinical presentation and in those patients with adverse outcomes (death or >30 days hospitalization). The increase in antibody levels is believed to be a response to an elevated viral load and may potentially be the culprit of uncontrolled systemic inflammatory response in critically ill patients [25,26]. IL-6 is known to induce defective angiogenesis via the activation of the Ang2 pathway [2]. Although this mechanism has been investigated in the context of malignancy and metastasis, the same principle applies to the influence of IL-6 on pro-inflammatory pathways during infection and sepsis [27]. The association between Ang1, Ang2, IgG antibody, and IL-6 in COVID-19 has been an interesting concept though we did not find any correlation between Ang2, IL-6, or IgG levels.

In summary, our findings document that decreased levels of Ang1 and elevated levels of Ang2 and IL-6 are hallmarks of COVID-19 infection and can be predictive of clinical outcomes. Our results agree with similar studies that clearly highlighted the link between elevated Ang2 levels in COVID-19 patients and ICU admission, mechanical ventilation, and death [17,28,29]. Thus, the data reported here suggest that Ang1 and Ang2 could be valid therapeutic targets for the development of novel treatment strategies. Further studies with a larger sample population would elucidate the exact pathophysiologic role of this inflammatory pathway in the progression of COVID-19. In addition, studies on the effects of these markers in patients with long COVID-19 syndrome would be an important future direction.

## Figures and Tables

**Figure 1 microorganisms-11-02940-f001:**
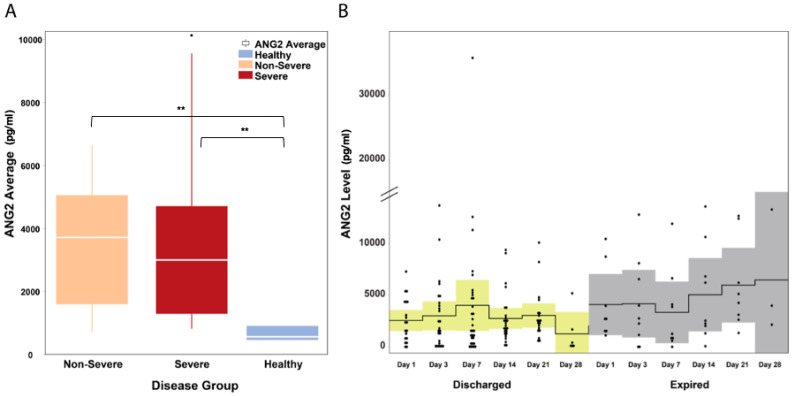
Ang2 levels in hospitalized COVID-19 patients. (**A**) Ang2 levels were significantly elevated (*P =* 0.0001) in all COVID-19 patients regardless of severity. The white horizontal line indicates the median, the box indicates the interquartile range, and the upper and lower whiskers represent values beyond 75% and 25%, respectively. (**B**) In patients who were discharged, Ang2 levels remained stable throughout hospitalization while in patients who died, there was an overall upward trend in Ang2 levels. The solid line indicates the mean and the colored areas shows the interquartile range. ** Statistically significant.

**Figure 2 microorganisms-11-02940-f002:**
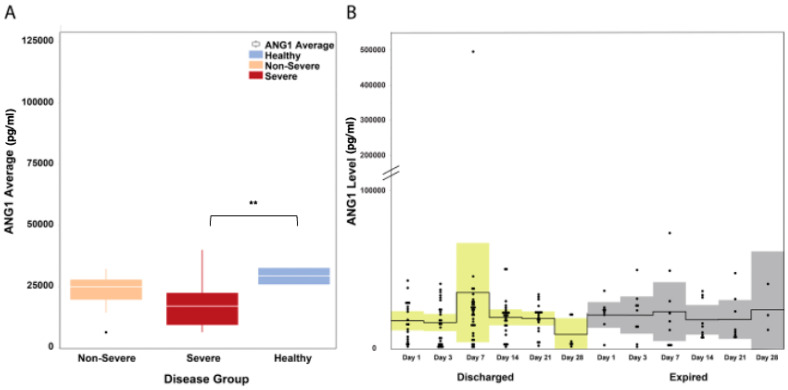
Ang1 levels in hospitalized COVID-19 patients. (**A**) Ang1 levels were mildly decreased in all COVID-19 patients regardless of severity. The white horizontal line indicates the median, the box indicates the interquartile range, and the upper and lower whiskers represent values beyond 75% and 25%, respectively. (**B**) A visual inspection of the time-course progression of Ang1 levels showed increasing levels on day 7 of illness of discharged patients, while levels remained relatively constant among patients who died. The solid line indicates the mean and the colored areas shows the interquartile range. ** Statistically significant.

**Figure 3 microorganisms-11-02940-f003:**
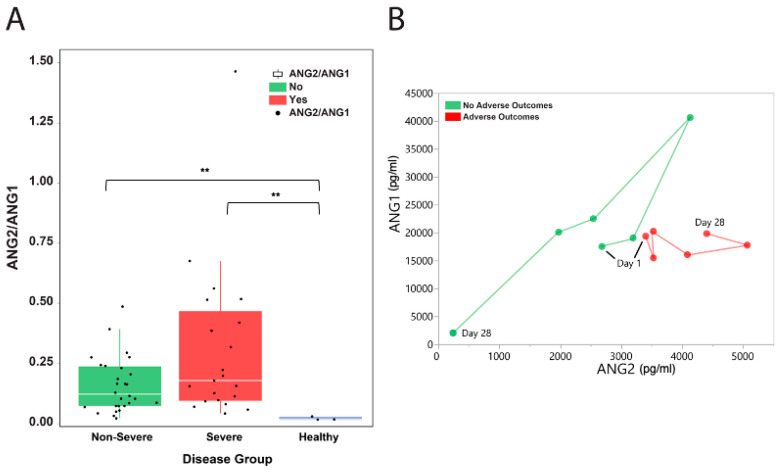
Ang2-to-Ang1 ratio in hospitalized COVID-19 patients. (**A**) The median Ang2-to-Ang1 ratio was elevated in COVID-19 patients (severe and non-severe) compared to healthy controls (median: 0.18 in non-severe versus 0.12 in severe, ANOVA *P* = 0.042). The white horizontal line indicates the median, the box indicates the interquartile range, and the upper and lower whiskers represent values beyond 75% and 25%, respectively. Asterisks indicate statistical significance. (**B**) Moving patient averages of Ang2 and Ang1 levels show the separation of patients based on the presence of adverse outcomes. ** Statistically significant.

**Figure 4 microorganisms-11-02940-f004:**
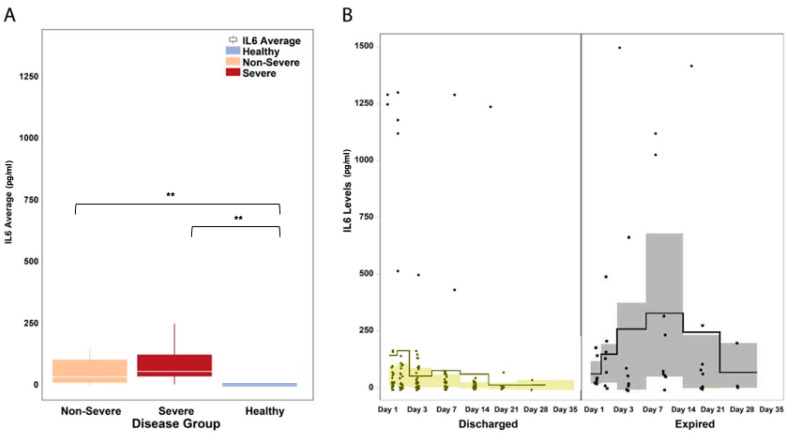
IL-6 levels in hospitalized COVID-19 patients. (**A**) The serum levels of IL-6 were significantly elevated in COVID-19 patients across disease severity (*P* = 0.047). IL-6 levels were significantly different between severe and non-severe patients (*P* = 0.001). The white horizontal line indicates the median, the box indicates the interquartile range, and the upper and lower whiskers represent values beyond 75% and 25%, respectively. (**B**) A visual inspection of the IL-6 trends shows that in patients who died from COVID-19 complications, IL-6 levels continued to rise during the first two weeks of illness and remained elevated until the third week. In patients who survived, IL-6 levels were observed to decrease after the first week of illness. The solid line indicates the mean and the colored areas shows the interquartile range. ** Statistically significant.

**Figure 5 microorganisms-11-02940-f005:**
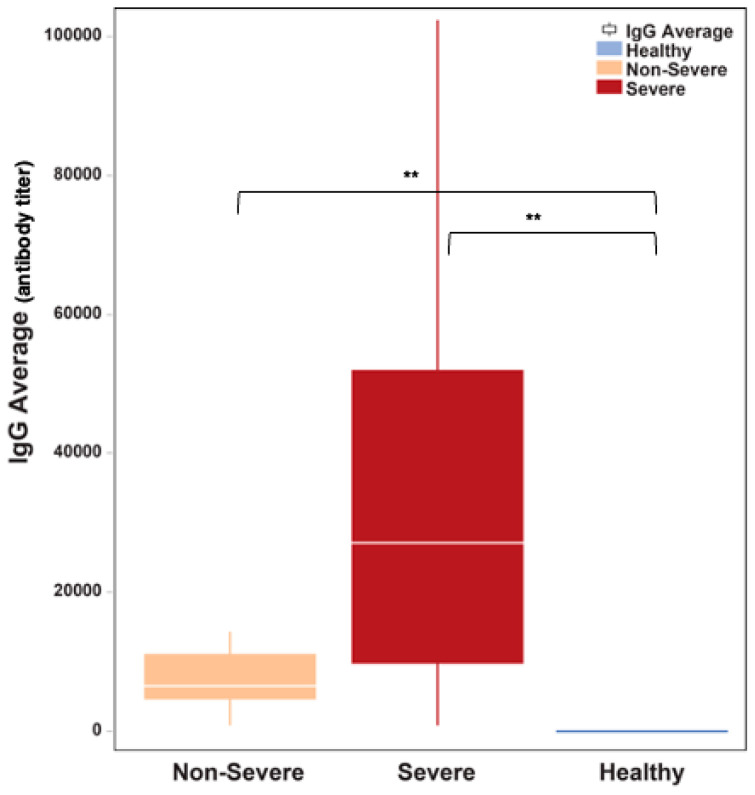
The amount of binding IgG antibodies was significantly elevated in COVID-19 patients who had severe disease relative to patients who manifested non-severity (*P* = 0.0001). The white horizontal line indicates the median the box indicates the interquartile range, and the upper and lower whiskers represent values beyond 75% and 25%, respectively. ** Statistically significant.

**Table 1 microorganisms-11-02940-t001:** Hospitalized COVID-19 patients’ characteristics.

Characteristics	Category	All (*n* = 49)	Non-Severe (*n* = 29)	Severe (*n* = 20)
Age	Median (IQR), years	63 (57–70)	63 (55–71)	63.5 (58.8–68.5)
Sex	Male	25 (51.0)	15 (51.7)	10 (50.0)
	Female	24 (49.0)	14 (48.3)	10 (50.0)
Race	Black	36 (73.5)	23 (79.3)	13 (65.0)
	Caucasian	8 (16.3)	5 (17.2)	3 (15.0)
	Asian	1 (2.0)	0 (0.0)	1 (5.0)
	Undisclosed	4 (8.2)	1 (3.4)	3 (15.0)
Co-morbid condition/s	Any condition	48 (98.0)	29 (100)	19 (95.0)
	HTN	38 (77.6)	25 (86.2)	13 (65.0)
	DM	32 (65.3)	21 (72.4)	11 (55.0)
	ESRD	14 (28.6)	12 (41.4)	2 (10.0)
	Cardiac disease (i.e., heart failure, arrhythmia)	14 (28.6)	8 (27.6)	6 (30.0)
	CNS disease (i.e., dementia, stroke)	10 (20.4)	6 (20.7)	4 (20.0)
	Lung disease (i.e., COPD, ILD)	6 (12.2)	1 (3.4)	5 (25.0)
	Cancer	2 (4.1)	2 (6.9)	0 (0.0)
	Others	19 (38.8)	12 (41.4)	7 (35.0)
Clinical Outcome	Discharged	37 (75.5)	28 (96.6)	9 (45.0)
	Expired	12 (24.5)	1 (3.4)	11 (55.0)
Length of hospitalization	Median (IQR), days	20 (9–28)	11 (8–22)	26 (19.3–31.3)

HTN—hypertension, DM—diabetes mellitus, ESRD—end-stage renal disease, CNS—central nervous system, COPD—chronic obstructive pulmonary disease, ILD—interstitial lung disease.

## Data Availability

The data presented in this study are available on request from the corresponding author. The data are not publicly available due to privacy.

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
