# Peer review of "The Angiopoietin Signaling Pathway Is Involved in Inflammatory Processes in Hospitalized COVID-19 Patients"

_microorganisms, 2023, doi:10.3390/microorganisms11122940_

Round 1

Reviewer 1 Report

Comments and Suggestions for Authors

Specific comments:

-The role of anti-RBD IgG data in the manuscript is unclear. Since authors do not provide any information on previous vaccinations and/or infections with SARS-CoV-2 for the studied patients, it is unclear how reliable fig. 5 data are. I think these limitations should be discussed in the text, and the point that presented IgG levels indeed reflect the status of SARS-CoV-2 infection at the moment of the study should be additionally supported.

-Figures: please show all statistically significant differences on the graphs. It is unclear which data are shown in panels A in all figures, since the patients were bled on multiple time points. Y axes should be clearly labeled in all graphs, units should be defined.

-Figure legends should avoid duplication of results description in the text. 

-Line 78: replace “levels Ang1” with “levels of Ang1”.

-2.1. Patient samples: indicate period (month/year) when patient samples were collected for the study. Also, as follows from the figures, serum samples were obtained on multiple days upon hospital admission – please indicate the specific days in this paragraph.

-Table 1: do authors have any information on previous vaccinations and infections with SARS-CoV-2? Numbers should be doublechecked (for example for cancer, 3 ≠ 2 + 0).

-Lines 133-134, recombinant RBD – is it of the original Wuhan strain (or WA1/2020)? 

-Lines 167-168: indicate IQR for the healthy group.

-Lines 176-177: not in agreement with fig. 1 legend (lines 184-185).

-Line 188, “in Covid-19 patients”: which group – severe or non-severe?

-Fig. 3A: the groups should be named as in other figures (severe and non-severe) to avoid confusion.

-Line 216: remove “also”.

-Lines 225-226: does not correspond to fig. 4A. The healthy group bar looks awkward and clearly differs from 0 pg/mL; the white horizontal line is invisible.

-Line 226, “median” – for which group? Or averaged non-severe + severe?

-Lines 230-232: switch the description for 2 groups to maintain the same order as in fig. 4B.

-Lines 245-246: this statement could be clearly supported by inclusion of panel B into fig. 5 showing the antibody dynamics (something similar to panels B in figures 1, 2 and 4).

-The Discussion section is missing an introductory paragraph.

-Lines 279-280, “in patients with severe clinical presentation and in those patients with adverse outcomes” – isn’t it the same group?

Comments on the Quality of English Language

The quality of English language is good.

Reviewer 2 Report

Comments and Suggestions for Authors

In this manuscript, the authors tried to investigate the levels of Ang1 and Ang2 in Covid‐19 patients during hospitalization, and correlate the levels according to severity of illness, anti‐SARS‐CoV‐2 IgG titers, interleukin IL‐6 level and clinical outcome by the ELISA kit and statistical analysis. The results sound interesting, while the following issues should be considered.

1.     There are various proteins involved in Covid‐19 patients, why the authors chosed the angiopoietin signaling pathway, especially Ang1, Ang2, and IL‐6? The details should be included in the Introduction section.

2.     The details of cases selection and curative effects should be included in the Materials and Methods section, especially how to determine the de‐identified samples.

3.     It is recommended to analyze the expression amount and expression activity of Ang1, Ang2, and IL‐6.

4.     The significance of this paper is not expounded sufficiently. The author needs to highlight this paper’s innovative contributions.

Comments on the Quality of English Language

5.     The grammar and spelling mistakes of this manuscript require the improvements. 

Round 2

Reviewer 2 Report

Comments and Suggestions for Authors

 The manuscript was adequately modified by the authors.